# The Association of Procalcitonin and C-Reactive Protein with Bacterial Infections Acquired during Intensive Care Unit Stay in COVID-19 Critically Ill Patients

**DOI:** 10.3390/antibiotics12101536

**Published:** 2023-10-12

**Authors:** Simone Campani, Marta Talamonti, Lorenzo Dall’Ara, Irene Coloretti, Ilenia Gatto, Emanuela Biagioni, Martina Tosi, Marianna Meschiari, Roberto Tonelli, Enrico Clini, Andrea Cossarizza, Giovanni Guaraldi, Cristina Mussini, Mario Sarti, Tommaso Trenti, Massimo Girardis

**Affiliations:** 1Intensive Care Unit, University Hospital of Modena, 41125 Modena, Italy; simo.campani.sc@gmail.com (S.C.); marty.talamonti@gmail.com (M.T.); lorenz.dallara@gmail.com (L.D.); irene.coloretti@gmail.com (I.C.); ilenia.gatto@gmail.com (I.G.); emanuela.biagioni@gmail.com (E.B.); tosimartina@gmail.com (M.T.); 2Infectious Disease Unit, University Hospital of Modena, 41125 Modena, Italy; meschiari.marianna@aou.mo.it (M.M.); giovanni.guaraldi@unimore.it (G.G.); cristina.mussini@unimore.it (C.M.); 3Respiratory Disease Unit, University Hospital of Modena, 41125 Modena, Italy; roberto.tonelli@me.com (R.T.); enrico.clini@unimore.it (E.C.); 4Immunology Laboratory, University of Modena and Reggio Emilia, 41125 Modena, Italy; andrea.cossarizza@unimore.it; 5Clinical Microbiology Laboratory, University of Modena and Reggio Emilia, 41125 Modena, Italy; sarti.mario@aou.mo.it; 6Diagnostic Hematology and Clinical Genomics Laboratory, Department of Laboratory Medicine and Pathology, Local Health Service and University Hospital of Modena, 41124 Modena, Italy; t.trenti@ausl.mo.it

**Keywords:** COVID-19, secondary infections, procalcitonin, C-reactive protein, intensive care unit

## Abstract

In COVID-19 patients, procalcitonin (PCT) and C-reactive protein (CRP) performance in identifying bacterial infections remains unclear. Our study aimed to evaluate the association of PCT and CRP with secondary infections acquired during ICU stay in critically ill COVID-19 patients. This observational study included adult patients admitted to three COVID-19 intensive care units (ICUs) from February 2020 to May 2022 with respiratory failure caused by SARS-CoV-2 infection and ICU stay ≥ 11 days. The values of PCT and CRP collected on the day of infection diagnosis were compared to those collected on day 11 after ICU admission, the median time for infection occurrence, in patients without secondary infection. The receiver operating characteristic curve (ROC) and multivariate logistic model were used to assess PCT and CRP association with secondary infections. Two hundred and seventy-nine patients were included, of whom 169 (60.6%) developed secondary infection after ICU admission. The PCT and CRP values observed on the day of the infection diagnosis were larger (*p* < 0.001) than those observed on day 11 after ICU admission in patients without secondary infections. The ROC analysis calculated an AUC of 0.744 (95%CI 0.685–0.803) and 0.754 (95%CI 0.695–0.812) for PCT and CRP, respectively. Multivariate logistic models showed that PCT ≥ 0.16 ng/mL and CRP ≥ 1.35 mg/dL were associated (*p* < 0.001) with infections acquired during ICU stay. Our results indicated that in COVID-19 patients, PCT and CRP values were associated with infections acquired during the ICU stay and can be used to support, together with clinical signs, rather than predict or rule out, the diagnosis of these infections.

## 1. Introduction

In critically ill COVID-19 patients, secondary bacterial infections and septic shock occurred more commonly than in other critically ill patients, increasing the risk of death by at least two-fold [1]. The high rate of secondary infections could be explained by many factors, including the deep immune dis-regulation induced by SARS-CoV-2 infection evolving from an initial excessive proinflammatory response to a profound immune dysfunction, exacerbated by treatment with steroids and other immunosuppressive medications [2,3]. Due to these reasons, the clinical signs and laboratory parameters commonly used to identify secondary infections were not helpful, and diagnosing secondary infections posed considerable challenges, especially in COVID-19 patients admitted to the ICU. 

The Food and Drug Administration, the WHO, and the Cochrane metanalysis endorse the use of procalcitonin (PCT) to guide antimicrobial stewardship in patients with severe respiratory infections, but these suggestions are derived from studies that did not include SARS-CoV2 infections [4,5,6]. In the context of viral infections like influenza or COVID-19, the utility of PCT and C-reactive protein (CRP) for predicting bacterial infections is largely debated, with numerous observational studies showing conflicting results [7]. The different cut-off values used, the setting of infection (community-acquired vs. hospital-acquired), and the prior utilization of antibiotics and other immune-modulating therapies contribute to the complexity of drawing definitive conclusions regarding the efficacy of these biomarkers during viral infections. In critically ill patients, using PCT and CRP to diagnose hospital-acquired pneumonia, including ventilator-acquired pneumonia, is not recommended by evidence-based guidelines [8,9], whereas their use is suggested for shortening the duration of antibiotic therapy. 

Most of the data available on COVID-19 patients analyze the ability of inflammatory biomarkers to identify infections at ICU admission [10,11], with very few studies exploring the performance of these biomarkers in infections acquired after admission. This large retrospective study in COVID-19 patients aims to evaluate whether PCT and CRP levels are associated and could help in the identification of secondary bacterial infections acquired during the ICU stay. 

## 2. Results

During the study period, 752 adult patients were admitted to our ICUs with respiratory failure associated with COVID-19, and 279 (37.1%) met the criteria to be included in the study, of whom 169 (60.6%) developed secondary infection after ICU admission. Patients with secondary infections during ICU stay received steroids, Tocilizumab, and invasive mechanical ventilation more frequently and showed larger ICU and hospital length of stays and mortalities. Secondary infections occurred in a median of 11 (IQR 6–16) days after ICU admission. Hospital-acquired pneumonia, including ventilator-associated pneumonia, accounted for nearly 90% of infections acquired during ICU stay (Table 1).

The PCT (median; IQR: 0.30, 0.12–1.00 ng/mL) and CRP (median; IQR: 3.2; 1.2–1.8 mg/dL) values observed the day of the infection diagnosis were larger (*p* < 0.001) than those (PCT median; IQR: 0.10, 0.10–0.12 ng/mL; CRP median; IQR: 0.4; 0.1–2.2 mg/dL) observed at day 11 after ICU admission in patients who did not develop infections (Figure 1, Appendix A). The ROC analysis calculated an AUC of 0.744 (95%CI 0.685–0.803) and 0.754 (95%CI 0.695–0.812) for PCT and CRP, respectively, with cut-off values of 0.16 (ng/mL) for PCT and 1.35 (mg/dL) for CRP providing a sensitivity >70% for the detection of secondary infection (Table 2 and Appendix A). Multivariate logistic models, adjusted for the use of steroids and Tocilizumab and the need for invasive mechanical ventilation, showed that PCT ≥ 0.16 ng/mL and CRP ≥ 1.35 mg/dL were associated (*p* < 0.001) with infection acquired during ICU stay (PCT: OR 6.71; 95%CI 3.78–11.90; CRP: OR 5.03; 95%CI 2.94–8.60). Invasive mechanical ventilation (OR 3.188; 95%CI 1.501–6.772; *p* < 0.001) and steroid therapy (OR 2.71; 95%CI 1.14 5.771; *p* = 0.024) but not Tocilizumab were also associated with infections acquired during ICU stay. The daily measurement of PCT and CRP from admission to day 14 of ICU stay are reported in Appendix A.

## 3. Materials and Methods

This observational retrospective cohort study was conducted in the three ICUs dedicated to COVID-19 patients of the Modena University Hospital and included consecutively admitted adult patients (>18 years) with respiratory failure and laboratory-confirmed SARS CoV-2 infection from February 2020 to May 2022. Patients with an ICU length of stay (LOS) <11 days, missed data, uncertain diagnosis of infection, limitation of care, or do not resuscitate orders were excluded from the study. The Ethics Committee of Area Vasta Nord Emilia Romagna approved the study, which deemed informed consent unnecessary because of the retrospective design (Approval Code: 0029747/20 Approval Date: 21 October 2020). We reported results following the indications from the Strengthening the Reporting of Observational Studies in Epidemiology (STROBE) guidelines [12].

SARS-CoV-2 infection was defined as a positive result of real-time reverse transcriptase–polymerase chain reaction (RT-PCR) assay targeted to nucleocapsid, envelope, and RNA-dependent RNA-polymerase genes of SARS-CoV-2 of nasopharyngeal swabs or lower respiratory tract specimens. All patients received standard ICU monitoring and supportive care according to disease severity. Pharmacologic therapy follows the international, national, and local guidelines [13,14,15]. 

For each patient, demographics, comorbidities, clinical parameters, laboratory data, and treatments were recorded according to internal protocol shared among the three ICUs. Specifically, PCT levels were obtained daily, and microbiological screening was performed at admission and repeated bi-weekly during ICU stay. In case of clinical suspicion of infection, blood and site-specific samples for microbiological examination were performed. The ICU and hospital length of stay and mortality were also collected. Infections were defined according to the international guidelines [16,17]. Bacterial pneumonia was defined as the presence of a new persistent infiltrate observed at the chest radiograph or computed tomography scan associated (at least one) with the worsening of oxygenation, purulent bronchial secretions, leukocytosis, and fever, and the presence of potentially pathogenic microorganisms in culture from tracheal aspirate and bronchoalveolar lavage. Bacteremia was defined as the presence of pathogenic microorganisms in the blood cultured from peripheral and/or central venous lines. The isolation of coagulase-negative Staphylococci in a single blood culture was not considered as bacteremia. An infectious disease specialist (MM) and a well-experienced intensivist (BM) have controlled and revised the clinical and microbiological data. Only infections that were microbiologically proven were considered. Infections occurring within 48 h since ICU admission were considered infections acquired before ICU and not included in the present analysis. 

To compare PCT and PCR values in patients with and without acquired infections, we included only patients with an ICU stay ≥11 days, the median time from ICU admission to the occurrence of secondary infections. The PCT and PCR values collected in patients with infection from one day before and one after the infection diagnosis of infection were compared to the values collected between 10 and 12 days after ICU admission in patients without infections. Non-parametric and χ^2^ tests were applied as appropriate to compare demographic, clinical and laboratory values, and outcomes between patients with and without secondary infections. The performance of PCT and PCR in detecting secondary infections was evaluated using the receiver operating characteristic curve (ROC), sensitivity, specificity, and negative and positive predictive value at different cut-off values obtained using ROC analysis or used in previous studies [9,16]. To assess the independent association of PCT and PCR levels with the occurrence of secondary infections, we used a multivariate logistic model, including variables with a *p* < 0.1 at unadjusted analysis. Continuous variables were expressed as medians and interquartile ranges, and categorical variables were expressed as percentages. The statistical analysis was conducted using the SPSS version 22.0 package (SPSS Inc., Chicago, IL, USA). The cut-off points of the P/C ratio in predicting mortality were estimated using receiver operator characteristic (ROC) curve analysis.

## 4. Discussion

This large retrospective analysis in COVID-19 patients with an ICU stay > 11 days indicated that PCT and CRP values were associated with developing secondary infections. For both biomarkers, the AUCs were higher than 0.7, which is the threshold required for an acceptable level of diagnostic accuracy [17]. The PCT and CRP values associated with infections were lower than those observed in non-COVID-19 patients [18,19,20]. Cut-off values of 0.16 ng/mL for PCT and 1.35 mg/dL for CRP may be used to support the diagnosis (PPV > 78%) but not for ruling out (NPV < 65%) secondary infections acquired during ICU stay. 

Several studies evaluated the performance of PCT and PCR in predicting bacterial infection in hospitalized COVID-19 patients, especially community-acquired bacterial co-infections [7]. The association between these biomarkers and co-infections at hospital or ICU admission has been evaluated in a large international study, including 4635 patients (7% with co-infections) admitted to 84 ICUs. The authors concluded that single measurements at ICU admission of PCT and PCR are not helpful (AUC for PCT 0.56 and CRP 0.54) in detecting infections, but baseline values of PCT < 0.3 ng/mL may be considered to rule out community-acquired bacterial co-infections (NPV 91.1%) [10]. Similarly, in 4076 COVID-19 patients (3% with co-infections) admitted to 55 Spanish ICUs over 20 months., a single PCT and PCR measurement at hospital admission showed low AUCs (0.57 and 0.60) in detecting co-infections. However, cut-off values of 0.12 ng/mL for PCT and 9.7 mg/dL for CRP provided high negative predictive values (97.5% and 98.2%) and, therefore, could be considered to rule out co-infections [10]. Another study (165 patients) demonstrated that a single value of PCT > 0.5 ng/mL in the first 72 h after ICU admission was not associated with an increased risk of bacterial pneumonia co-infection, with a sensitivity of 26% and a negative predictive value of 73% [21]. A recent meta-analysis, including five studies involving 2775 patients, confirmed the poor predictive value of PCT for co-infections in patients with COVID-19, and that lower PCT levels indicate a decreased probability of having a co-infection [22].

To our knowledge, only a few studies have evaluated in COVID-19 patients the relationship between infections acquired during ICU stay and inflammatory biomarkers. A small retrospective study on 52 ICU and 47 no-ICU COVID-19 patients indicated that PCT and CRP may be useful in identifying secondary bacterial infections and guiding the use of antibiotics. The best cut-off values were identified as 0.55 ng/mL for PCT and 17.3 mg/dL for CRP, providing high (>76%) sensitivity and specificity and very high NPV (>90%) in predicting secondary infections [23]. These performances were higher than those calculated in our cohort at the same cut-off values and even using the best thresholds calculated using ROC analysis (see Table 2). The difference could be explained by the exclusion from our study’s analysis of co-infections at ICU admission and the anti-inflammatory treatments received. Most of our patients received steroids for at least ten days, and around 80% also had one dose of Tocilizumab, whereas the use of steroids or other immune therapies has not been reported in the study by Pink et al. [23]. In non-COVID-19 patients with sepsis, using steroids and Tocilizumab seems to induce a reduction in CRP and PCT levels and less response to infections, which is more pronounced on CRP levels [24,25]. A single-center study in COVID-19 patients admitted to the ICU (71 with vs. 79 without secondary infections) demonstrated that steroids and Tocilizumab, alone or combined, cause the suppression of PCT and CRP production, with a considerable decrease in the relationship between these biomarkers and secondary infections [26]. A meta-analysis of nine observational trials reported a median reduction of 0.67 ng/dL and 10.6 mg/dL for PCT and CRP, respectively, after the administration of Tocilizumab in COVID-19 patients [27]. Our cohort’s significant use of steroids and Tocilizumab may justify the low predictive negative values at different cut-off values. Unfortunately, we cannot further analyze this aspect because of our study’s low number of patients not receiving anti-inflammatory drugs. 

Different from our results, a retrospective single-center study including COVID-19 patients admitted to the ICU (143 with and 99 without infection) in the first wave concluded that PCT values were not associated with positive cultures of respiratory and urine tracts, whereas they were increased in patients with bacteremia [28]. Indeed, the data showed higher values of PCT values in positive cultures compared to negative cultures, even in sputum (median 1.65 vs. 0 ng/mL; *p* < 0.01) and urine cultures (median 0.95 vs. 0.52 ng/mL; *p* < 0.01). Unfortunately, the authors did not report cut-off values and analysis of PCT performance. Similarly, Harte et al. [29] did not observe any relationship between PCT and infections at admission and during ICU stay, but they did not provide specific data on PCT on the day of infection and the percentage of patients receiving steroids and/or other immunotherapies. The lack of proper analysis on PCT and CRP performance and information on anti-inflammatory therapies make the above studies difficult to compare to other studies and our results.

Interestingly, a small observational study in COVID-19 patients admitted to the ICU during the first wave reported a remarkable association between PCT increase (>50% of the previous value) and the development of secondary bacterial pneumonia. In contrast, no association was observed between CRP and blood cell increase [30]. Unfortunately, using steroids and/or Tocilizumab that can justify the difference observed between biomarkers is not reported.

In our study, the AUC (0.744 and 0.754; *p* = 0.7926) and the diagnostic performance of the different levels of PCT and CRP are comparable in detecting acquired infections during ICU stay. Therefore, for this purpose, the use of both biomarkers is not indicated, and it may be advisable to use only the least expensive of the two. 

Our study is the largest experience reporting the changes in PCT and CRP related to secondary bacterial infections acquired during the ICU stay of critically ill COVID-19 patients. Moreover, using data obtained after ten days of ICU stay in non-infected patients makes robust comparisons between infected and non-infected patients, considering the potential effects of COVID-19 and specific therapies on immune response and inflammatory biomarkers. The retrospective design and the use of only microbiologically proven infections are the significant limitations of our study. Therefore, we decided to avoid any analysis of the effects of PCT and CRP values on the clinical decision regarding antibiotic initiation, de-scalation, and withdrawal. 

## 5. Conclusions

To sum up, as observed for non-COVID-19 patients, our data and literature indicate an association between inflammatory biomarkers and infection acquired during ICU, especially pneumonia. The use of steroids and other immune therapies may blunt the inflammatory response to infection and, thus, the cut-off values commonly used in non-COVID-19 patients, such as 0.5 ng/mL for PCT and 10 mg/dL, have low predictive positive values but can be helpful to rule out bacterial infections at hospital or ICU admission, particularly community-acquired bacterial pneumonia. In critically ill COVID-19 patients receiving therapies to modulate inflammatory response, low cut-off values of PCT (<0.2 ng/mL) and PCR (<2 mg/dL) show an acceptable performance that can be used to support, together with other clinical signs, rather than predict or rule out, the diagnosis of secondary infection acquired during ICU stay admission.

## Figures and Tables

**Figure 1 antibiotics-12-01536-f001:**
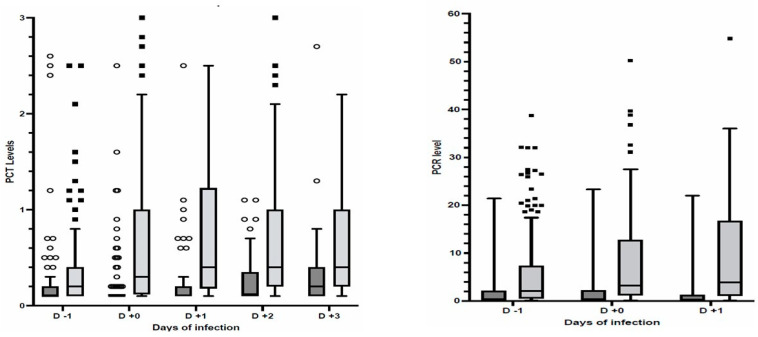
Scatter box of PCT (left panel) and CRP (right panel) levels before and after infection diagnosis or day 11 after ICU admission in patients without infections. Patients with secondary infections (dark grey); patients without infections (light grey); PCT in ng/mL; CRP in mg/dL; Outliers (circles and squares) are also reported. *p* < 0.001 between patients with and without secondary infections.

**Table 1 antibiotics-12-01536-t001:** Demographic characteristics, severity scores, and laboratory parameters at ICU admission, treatments received during ICU stay, site of infections, length of stay, and mortality in all patients included in the analysis, subdivided into patients with and without secondary infections during ICU stay.

	All Patients (*n* = 279)	NO Infection (*n* = 110)	YES Infection (*n* = 169)	*p* Value
Age (years; median, IQR)	67 (60–73)	67 (58–73)	67 (61–72)	0.946
Sex, male (*n*, %)	211 (76)	93 (79)	118 (74.7)	0.576
BMI (median, IQR)	29 (26–33)	29 (26–32)	29 (26–33)	0.710
SAPSII (median, IQR)	35 (30–41)	35 (29–42)	35 (32–40)	0.349
CRP admission (mg/dL; median-IQR)	6.5 (2.4–17.0)	7.2 (2.7–15.5)	6.3 (2.1–17.7)	0.356
PCT baseline (ng/mL; median, IQR)	0.2 (0.1–0.6)	0.2 (0.1–0.6)	0.2 (0.12–0.6)	0.545
Steroids administration (*n*, %)	270 (97)	115 (95)	155 (98)	0.068
Tocilizumab (*n*, %)	219 (78)	86 (71)	133 (84)	0.029
Invasive mechanical ventilation (*n*, %)	249 (89)	97 (80)	152 (96)	<0.001
Site of secondary infection (*n*, %)				
Lung		-	143 (90.1)	-
Blood Stream		-	75 (47.3)	-
Other sites		-	44 (26.2)	-
ICU LOS (days; median, IQR)	18 (11–35)	13 (10–19)	28 (15–45)	<0.001
Hospital LOS (days; median, IQR)	33 (21–49)	27 (20–37)	38 (24–59)	<0.001
ICU mortality (*n*, %)	119 (43)	29 (26.2)	90 (54.1)	<0.001
Hospital mortality (*n*, %)	130 (47)	34 (31)	96 (57)	<0.001

BMI: body mass index; SAPSII: Simplified Acute Physiology Score II; CRP: C-reactive protein; PCT: procalcitonin; ICU: intensive care unit; LOS: length of stay.

**Table 2 antibiotics-12-01536-t002:** Sensitivity, specificity, positive and negative predictive value of different PCT and CRP cut-off values in detecting secondary bacterial infections in patients admitted to intensive care unit. Data are in % (95 CI).

	Sensitivity	Specificity	Positive Predictive Value	Negative Predictive Value
PCT (ng/mL)				
≥0.1	100.0 (97.8–100.0)	0.0% (0.0–3.3)	61.5% (61.5–61.5)	-
≥0.16	71.0 (63.5–77.7)	76.4% (67.3–83.9)	82.2% (75.0–88.0)	63.1% (54.3–71.3)
≥0.25.	60.9 (53.2–68.3)	83.6% (75.4–90.0)	85.2% (77.5–91.0)	58.2% (50.1–66.0)
≥0.5	43.2 (35.6–51.0)	86.4% (78.5–92.2)	83.0% (73.5–90.2)	49.7% (42.4–57.1)
≥1	26.0 (19.6–33.3)	93.6% (87.3–97.4)	86.3% (73.8–94.3)	45.2% (38.6–51.9)
CRP (mg/dL)				
≥1	77.5% (70.5–83.6)	63.6% (53.9–72.6)	76.6% (69.6–82.8)	64.8% (55.0–73.7)
≥1.35	72.2% (64.8–78.8)	68.2% (58.6–76.7)	78.4% (73.1–82.9)	60.1% (53.8–66.9)
≥2	62.1% (54.4–69.5)	73.6% (64.4–81.6)	79.0% (73.0–84.0)	54.9% (49.3–60.3)
≥5	38.5% (31.1–46.2)	87.3% (79.6–92.9)	82.8% (74.1–89.1)	47.0% (43.6–50.5)
≥10	26.6% (20.1–34.0)	95.5% (89.7–98.5)	90.0% (78.2–96.7)	45.8% (38.2–52.5)

## Data Availability

The data presented in this study are available on request from the corresponding author.

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
