# Peer review of "The Association of Procalcitonin and C-Reactive Protein with Bacterial Infections Acquired during Intensive Care Unit Stay in COVID-19 Critically Ill Patients"

_antibiotics, 2023, doi:10.3390/antibiotics12101536_

Round 1

Reviewer 1 Report

An interesting manuscript analyzing the association of procalcitonin and C-reactive protein with bacterial infections acquired during ICU stay in COVID-19 critically ill patients is presented.

I have no significant comments on the article. The introduction is concise, the methodology is adequate, and the discussion corresponds to the obtained results and other published articles on this topic.

I have two minor comments.

I would consider it appropriate if the authors specifically stated the clinical criteria for establishing the diagnosis of pneumonia and bloodstream infection.

I recommend specifying the type of pneumonia more precisely. It is obvious that these were hospital-acquired pneumonias, but this should be specified in the text.

Author Response

  1. I would consider it appropriate if the authors specifically stated the clinical criteria for establishing the diagnosis of pneumonia and bloodstream infection.

We added the specifications required in the methods.

  1. I recommend specifying the type of pneumonia more precisely. It is obvious that these were hospital-acquired pneumonias, but this should be specified in the text.

In the table, the word pneumonia has been replaced by lung as indicated by reviewer 2.

Reviewer 2 Report

On line 56, the statement "...the time of infection (community-acquired vs  hospital-acquired)" should be reworded. Time (such as hour of the day) is unrelated to the setting of infection (community or hospital).

Line 77: state the molecular targets for your SARS-CoV-2 PCR.

Reword the statement "Only infections microbiologically proven infection was considered" on line 90 to "Only infections that were microbiologically proven were considered."

Since "microbiologically proven infections" are absolutely critical for the study, some elaboration on what this means has to be provided in the materials and methods section. 

In Table 1, the categories "NO infection" and "YES infection" should be reworded. For example, authors can consider using "Infection absent" and "Infection present." 

Still regarding Table 1, "pneumonia" is not a site of infection but a type of infection. Use "lungs" for site of infection. Likewise "other infections" should be replaced with "other sites."

"LOS" has to be spelled out in full in the footer of Table 1.

On line 199, the statement "In COVID-19 patients, a single center study admitted to ICU..." is grammatically erroneous. It implies that rather than patients, it was the ICU which was admitted.

On line 243, "community bacterial pneumonia" should more accurately be stated as "community-acquired bacterial pneumonia."

The manuscript has many grammatical errors (some of which I have already pointed out above). Therefore, proofreading by an English language expert is mandatory.

Author Response

  1. On line 56, the statement "...the time of infection (community-acquired vs hospital-acquired)" should be reworded. Time (such as hour of the day) is unrelated to the setting of infection (community or hospital).

Modified as indicated.

  1. Line 77: state the molecular targets for your SARS-CoV-2 PCR.

Inserted as requested.

  1. Reword the statement "Only infections microbiologically proven infection was considered" on line 90 to "Only infections that were microbiologically proven were considered."

Modified as indicated.

  1. Since "microbiologically proven infections" are absolutely critical for the study, some elaboration on what this means has to be provided in the materials and methods section.

As requested by reviewer 1, we better specified the infection definitions.

  1. In Table 1, the categories "NO infection" and "YES infection" should be reworded. For example, authors can consider using "Infection absent" and "Infection present."

Modified as indicated.

  1. Still regarding Table 1, "pneumonia" is not a site of infection but a type of infection. Use "lungs" for site of infection. Likewise "other infections" should be replaced with "other sites."

Modified as indicated.

  1. "LOS" has to be spelled out in full in the footer of Table 1.

Modified as indicated

  1. On line 199, the statement "In COVID-19 patients, a single center study admitted to ICU..." is grammatically erroneous. It implies that rather than patients, it was the ICU which was admitted.

Modified as indicated

  1. On line 243, "community bacterial pneumonia" should more accurately be stated as "community-acquired bacterial pneumonia."

Modified as indicated

  1. Comments on the Quality of English Language: The manuscript has many grammatical errors (some of which I have already pointed out above). Therefore, proofreading by an English language expert is mandatory.

The manuscript has been checked by Paperpal and Grammary.